# The Importance of Detecting, Quantifying, and Characterizing Exosomes as a New Diagnostic/Prognostic Approach for Tumor Patients

**DOI:** 10.3390/cancers15112878

**Published:** 2023-05-23

**Authors:** Mariantonia Logozzi, Nicola Salvatore Orefice, Rossella Di Raimo, Davide Mizzoni, Stefano Fais

**Affiliations:** 1Department of Oncology and Molecular Medicine, Istituto Superiore di Sanità, 00161 Rome, Italy; mariantonia.logozzi@iss.it; 2Department of Pharmacology, Feinberg School of Medicine, Northwestern University, Chicago, IL 60611, USA or nicola.orefice@northwestern.edu; 3ExoLab Italia, Tecnopolo d’Abruzzo, 67100 L’Aquila, Italy; rossella@exolabitalia.com (R.D.R.); davide@exolabitalia.com (D.M.)

**Keywords:** extracellular vesicles, exosomes, tumors, biomarkers, methodology, body fluids, plasma

## Abstract

**Simple Summary:**

Clinical oncology urgently needs more specific and helpful new biomarkers to improve the diagnosis and prognosis of cancer. Research of the last decade proposes extracellular vesicles, particularly exosomes, as a natural source of new biomarkers; since tumors massively release them, they circulate through the body and can be detected and characterized in plasma samples of tumor patients. After a decade of up-and-coming pre-clinical research, the results of the few clinical studies have provided some exciting data supporting the use of exosomes, at least in the follow-up of tumor patients. However, the most convincing data have taught us that, on the one hand, circulating exosomes deliver known tumor markers, such as PSA; on the other hand, the exosome plasmatic levels in tumor patients consistently exceed those of normal controls. This information will be extremely useful in the clinical management of tumor patients.

**Abstract:**

Exosomes are extracellular vesicles (EVs) of nanometric size studied for their role in tumor pathogenesis and progression and as a new source of tumor biomarkers. The clinical studies have provided encouraging but probably unexpected results, including the exosome plasmatic levels’ clinical relevance and well-known biomarkers’ overexpression on the circulating EVs. The technical approach to obtaining EVs includes methods to physically purify EVs and characterize EVs, such as Nanosight Tracking Analysis (NTA), immunocapture-based ELISA, and nano-scale flow cytometry. Based on the above approaches, some clinical investigations have been performed on patients with different tumors, providing exciting and promising results. Here we emphasize data showing that exosome plasmatic levels are consistently higher in tumor patients than in controls and that plasmatic exosomes express well-known tumor markers (e.g., PSA and CEA), proteins with enzymatic activity, and nucleic acids. However, we also know that tumor microenvironment acidity is a key factor in influencing both the amount and the characteristics of the exosome released by tumor cells. In fact, acidity significantly increases exosome release by tumor cells, which correlates with the number of exosomes that circulate through the body of a tumor patient.

## 1. Introduction

Virtually any cell, during its lifespan (from embryonic development to senescence), releases extracellular vesicles (EVs). EVs range in size from 30 nm to 1 μm, and the size distinguishes microvesicles (from 200 nm to 1 μm) from exosomes (from 30 nm to 150 nm) under both physiological and pathological conditions [1,2,3,4,5,6].

Exosome generation processes include a membrane shedding-like phenomenon (for microvesicles) and multivesicular body (MVB) formation (for exosomes) [5,6,7,8]. Other mechanisms cannot be excluded and are currently under investigation worldwide. EVs, particularly nanovesicles (exosomes), are a natural delivery system for a wide array of substances. Exosomes travel the body through both hematic and lymphatic circulations. Between the molecules that may be detected in exosome preparations, there are housekeeping proteins, including tetraspanins (i.e., CD63, CD9, and CD81), heat shock proteins (such as HSP-70), members of the Rab family, as well as other proteins, including Tsg101 and Alix. These markers have been used to characterize and quantify exosomes [9,10]. However, exosomes, during their formation, involve internal cell structures and the plasma membrane; this may lead to the acquisition of markers of the cellular source [5,6,7,8].

Overall, exosomes purified from body fluids may contain typical tags that help distinguish exosomes from other particles and markers indicating the cellular source and often the body compartment from which they are released. The above reasons make EVs, particularly exosomes, a potential source of disease biomarkers with possible use as a liquid biopsy in clinical oncology. In addition, exosomes have been shown to contain a series of nucleic acids, including DNAs, mRNAs, and miRNAs, that may represent an additional source of disease biomarkers [5,6,11]. However, exosomes also remove unnecessary molecules poorly degraded by the lysosomal system [12,13], thus emphasizing the broad and complex function of these nanovesicles in our body [1,4,5,6,12,13,14,15,16,17,18,19,20].

Exosomes can be found in many biological fluids, including blood, urine, saliva, stools, cerebrospinal, epididymal, amniotic, serous fluids (including pleural, pericardial, and peritoneal fluids), bronchoalveolar lavage fluid, synovial fluid, and breast milk [6,21,22,23,24,25,26]. Exosomes are released in a paracrine way within tissues, from where they are spilled into the bloodstream, often ending in tissues of body compartments far from the production site. For example, scientific evidence has shown that exosomes containing a reporter gene are released from a tumor, found in the blood, and end in the germ line of the gonads, with the potential to transfer the acquired genetic material to the progeny [27]. It is, therefore, conceivable that exosomes may well participate in the continuous genome remodeling that occurs in our body. The matter of fact is that exosomes are considered a natural source of disease biomarkers [5,6,11,25,26,27,28,29,30,31,32,33,34,35,36,37]. A series of exciting molecules have been identified in the plasma of both patients and healthy donors [5,6,7,11,29,32,38]. Clinical studies, while still very few as compared to pre-clinical information, are providing exciting information while often not entirely fitting with the aim of the studies [11,39,40,41], challenging the use of these data in clinical settings. The future goal of translational oncology is and will be to define the molecules’ cargo of body fluid-derived exosomes in tumor patients, also based on the evidence that tumor-released exosomes are involved in both tumor progression and metastasis [1,4,11,15]. Some unexpected but interesting findings propose the simple measurement of exosome plasmatic levels as a key prognostic value [41,42]. The clinical data show that, independently from the cancer histology (i.e., melanoma, prostate cancer, or oral cancers) and the technique used in the experimental protocol (e.g., immunocapture ELISA, nanoparticle tracking analysis (NTA), or nanoscale flow cytometry), patients displayed higher plasmatic exosome levels as compared to healthy donors [24,41,42,43]. Other interesting issues are the expression on plasmatic exosomes from tumor patients of acknowledged tumor markers (e.g., PSA and CEA) [44,45] and a series of surrogate tumor markers (e.g., Cav-1, HSP60, and carbonic anhydrase, such as CA IX) [24,43,46,47]. This review will introduce and discuss these issues to propose the best use of exosomes in clinical oncology.

## 2. A Technical Insight

A general discussion is given of the techniques used to purify and characterize exosomes from patient samples [5]. Currently, Nanoparticle Tracking Analysis (NTA) allows the determination of the number and size of the obtained EVs from either cell culture supernatant or body fluids. NTA acquires the Brownian movement of nanoparticles in a liquid suspension, analyzing the EVs’ concentration and size distribution in the sample. This is based on a single particle analysis with a serial correlation with the particle size [47,48]. The NTA analysis covers a broad range of particle sizes, ranging from 30 nm to 400–500 nm, thus distinguishing nanovesicles from microvesicles. NTA is, to date, considered the most reliable technique to analyze a mixed population of submicroscopical vesicles in human body fluids.

A preliminary analysis that might be performed on an EVS sample is transmission electron microscopy (TEM). While not allowing a quantitative evaluation, TEM is an integral approach to verifying whether samples under investigation contain submicroscopical vesicles and whether a round shape and the typical bilayer membrane are maintained after repeated centrifugation and ultracentrifugation. Moreover, vesicles may be phenotyped by immuno-TEM using immuno-gold-labeled antibodies. A disadvantage of TEM is that the samples undergo sequential rounds of fixing and dehydration before analysis, thus potentially inducing morphological damage [47,48]. However, it is advisable to evaluate exosomes by TEM analysis.

A rough evaluation of exosomes may also be performed by measuring the amount and type of exosomal proteins present in the sample. The last accepted guidelines (MISEV2018) have agreed on the following points that are required to establish that the sample under investigation contains exosomes: (i) enrichment in at least one transmembrane protein associated with the exosomal plasma membrane (e.g., tetraspanins CD9, CD63, CD81); (ii) enrichment in cytosolic proteins (e.g., TSG101, ALIX) [5,49]. The most commonly used techniques allowing this analysis are (i) Western blot, which is only a semi-quantitative approach not valid for the study of clinical samples. Moreover, it is expensive in terms of both the volumes required for the analysis and the time needed to obtain the results; it is undeniably a qualitative analysis, allowing the detection of many proteins at the same time; and (ii) flow cytometry allows simultaneous analysis of phenotyping (through labeling with fluorescent antibodies) and physical parameters (e.g., size and structure of particles). However, conventional cytometers could underestimate particles smaller than 300 nm, and a new generation of flow cytometers has been provided with both multi-angle lasers to improve particle resolution [50,51,52] and nanoscale equipment to include analysis of nanosized particles, also called nanoscale-flow cytometry, recently used in clinical studies [44,53].

A technical approach that allows us to simultaneously provide quantitative and qualitative data is the immunocapture-based ELISA. It was shown for the first time that immunocapture-based ELISA exosomes could be quantified and characterized from either cell culture supernatants or human plasma [24]. This technique was exploited in clinical investigations, including melanoma, prostate, and oral cancer patients [24,43,44]. This approach allows the analysis of the whole EV population, including exosomes. Fluorescence Activated Cell Sorter (FACS), while equipped with nanoscale flow cytometry, does not allow a broad spectrum of analysis or simultaneous analysis of different samples. Immunocapture-based ELISA looks ideal for this purpose since it will enable the detection and quantification of both exosome-specific antigens and tumor antigens on EVs isolated from small quantities of plasma simultaneously [24,43,44,53]. Recent data support the high level of versatility of the technique, with the identification of a series of housekeeping proteins, such as Rab5b, CD81, and CD63, and tumor-specific markers, such as PSA, but also surrogate tumor markers, such as Cav-1 and carbonic anhydrase [24,43,44,46,47,53].

Furthermore, this approach has been recently reported for characterizing urinary exosomes [25], thus representing a new approach for the follow-up of patients affected by urinary tract cancers. However, the goal will be to implement immunocapture-based ELISA with other methods, such as nanoscale flow cytometry (NFC) and NTA, as proposed in prostate cancer patients [41]. In the above study, statistical analysis of the results showed that immunocapture-based ELISA allows exosomal PSA detection and discriminates prostate cancer patients from both healthy subjects and benign prostate hypertrophy (BPH) patients with significantly higher sensitivity and specificity than serum PSA. Moreover, immunocapture-based ELISA allows for quantifying and characterizing several clinical samples simultaneously and in a broader population of EVs compared to nanoscale flow cytometry [53,54,55].

## 3. A Role of Exosomes in Cancer: From Preclinical to Clinical Data

Scientific evidence is accumulating that exosomes have a crucial role in tumor metastasis, passing through either the generation of a metastatic niche or a tumor-like transformation of mesenchymal stem cells in organs that are targets of metastasis [4,15,56,57,58]. However, the acidic pH of the tumor microenvironment plays a determinant role in at least three essential features: (i) the increased exosome release by tumor cells; (ii) determining the exosome cargo, including some tumor biomarkers [2,46,53]; and (iii) it is associated with a reduced size as compared to the heterogeneous size of those released at physiological pH [2,53]. The increased exosome release in acidic conditions correlates to the high plasmatic exosome levels compared to controls [44,53]. The reason why tumor cells increase the release of exosomes in acidic conditions may be related to the attempt to eliminate toxic molecules that tend to accumulate in the tumor microenvironment; the molecules to stop include antitumor drugs such as cisplatin [59]. This is further supported by the observation that antitumor medications contained in the exosomes released by tumors are in their native/active form, thus potentially being released into the bloodstream and getting into unaffected organs, contributing to the heavy side effects that sadly often occur in cancer patients. Between the molecules delivered by tumor exosomes, there are ion transporters (e.g., CAIX) that, together, are significantly increased in exosomes released in acidic conditions and conserve their full enzymatic function [46]. The CA has also been shown in the plasmatic exosomes of cancer patients; the same plasmatic exosomes have shown increased acidity compared to healthy subjects [47].

Another hurdle was the claim for the specificity of some markers identified on circulating exosomes of tumor patients that turned out not to be so specific for a given tumor. One example is glypican-1, which has been proposed as a specific marker of pancreatic cancer but also showed a high expression level in exosome purification from other cancers [56]. Too often, the specificity of an exosome-related tumor biomarker was not tested by comparing different cancer patients [60].

## 4. Exosomes Deliver Enzymatic Activity

One of the most effective mechanisms by which exosomes may up-load their content into target cells is the fusion between their membrane and the plasma membrane of a target cell [61]. Through the above mechanism, exosomes released by a primary tumor may contribute to the metastatic process once they get to a metastatic organ via the bloodstream [15,58]. This is further supported by a recent report showing that exosomes obtained from cancer patients’ plasma deliver proteins and molecules with evident enzymatic activity and an intraluminal pH suitable for enzyme activation [47]. Notably, it was also shown that in vitro, the acidic condition increases the expression of exosomes and proteins with enzymatic activity, such as carbonic anhydrase [46]. This information, on the one hand, further highlights the importance of exosomes as a natural delivery system for a broad array of molecules; on the other hand, it suggests that the research of disease biomarkers should also be directed to functional molecules rather than the mere expression of a protein.

## 5. Exosomes Deliver Nucleic Acids

At the time, exosomes were considered vesicles released by the cells with a significant commitment to scavenging cells from either toxic or unwanted material. Of course, this remains a function of extracellular vesicles, as witnessed by EVs in the stools and urine [6]. However, the discovery that EVs deliver nucleic acids has changed how these vesicles have been considered [19]. It has been shown that EVs, purified from either cell culture supernatant or human body fluids, contain mRNA, miRNA, long non-coding RNA (lncRNA), and DNA [11,62,63]. Most clinical studies reporting the nucleic acid cargo of body fluid-derived exosomes have been performed in tumor patients. The results suggest significant differences exist between tumor patients and healthy individuals, particularly in exosomal miRNA composition [32,64,65,66,67,68,69,70,71]. Currently, there is some inconsistency primarily due to technical and analytical issues, which too often create inhomogeneity between the samples, which in turn affects miRNA’s yield, integrity, and purity [5]. One important issue is the evidence that miRNAs are not always associated with exosomes, often being associated with either RNA-binding proteins (e.g., Argonaute 2) or lipoproteins (e.g., HDL and LDL) [5,65,66]. More recently, a commercially available isolation kit (MACS Exosome Isolation Kit, Miltenyi Biotec, Germany) is starting to be exploited to obtain a more purified exosome population, thus providing a more certain exosome-associated miRNA yield [72]. Comparably to the MACS method, immunocapture-based exosome purification may greatly help in obtaining exosomes from the ultracentrifuged material using antibodies directed against the proteins that are overexpressed on the exosome membrane (e.g., CD9, CD63, CD81, ALIX). The same approach may be exploited using plastic wells and magnetic beads as primary substrates [73]. This approach allows us to obtain a highly enriched exosome preparation, thus analyzing only the vesicles captured by the antibodies in terms of characterization of either miRNAs or RNAs present in the immunocaptured material. The immunocapture-based methodology has also been described and used in clinical trials [53,54,55]. However, it needs to be extensively exploited in analyzing the presence of exosome-associated nucleic acids in clinical samples using different approaches [74]. Another interesting area is related to the analysis of the presence of genomic DNA mutations in exosomes purified from clinical samples. DNA mutations are involved in many tumor advantages, most notably resistance to therapies, and represent a potential tumor biomarker [71]. Detecting exosomal DNA in clinical samples is receiving a large consensus in cancer patients [75,76,77,78] and other diseases, including viral-related pathological conditions [79]. In addition, recent reports have shown that exosomes purified and concentrated from body fluids, such as ascites, may express high levels of protein glycosylation [80]. While the data reporting critical roles of exosome associated RNAs is becoming bulky, we need more convincing evidence that they may represent helpful and reliable tumor biomarkers to be diffusely used in oncology laboratories worldwide. Therefore, it appears mandatory that it should need central management of the available data to get to a conclusive analysis.

## 6. Conclusions

To date, we have considerable data supporting the use of exosomes and EVs for the clinical management of tumor patients (Table 1). However, of course, it needs clinical validation to be considered an accurate diagnostic/prognostic tool in clinical oncology. What was an exciting hypothesis for the scientists involved in the field a decade ago is now scientific evidence that exosomes are a source of new biomarkers. However, while the discovery of new biomarkers still needs time to be translated into the clinic, some unexpected findings promise to need a shorter path to clinical use: (1) The evaluation of the number of circulating exosomes that are proven to be higher in patients with cancer as compared to healthy controls; (2) Plasmatic exosomes hyperexpress known tumor biomarkers (e.g., PSA, CEA).

Additional information is that plasmatic exosomes are smaller in tumor patients than in healthy and diseased controls and more acidic in tumor patients than controls. Thus, quantifying and characterizing exosomes in human body fluids represents a new tool for clinical oncologists and a non-invasive diagnostic/prognostic approach.

We have three methods that, when implemented, may offer a solid approach to using these methods together to quantify and characterize exosomes: Nanoparticle Tracking Analysis (NTA), immunocapture-based ELISA, and nanoscale flow cytometry (NFC). Using all these methodologies to describe exosome purification in clinical samples may represent a real advance in the clinical management of tumor patients. Another interesting approach is to use immunocapture of exosomes to optimize the detection of tumor biomarkers, particularly in detecting and validating tumor-specific miRNA. Possible future directions could be: (i) to identify physical-chemical properties of exosomes associated with some tumor phenotypes (e.g., intraluminal pH); (ii) to include the expression of active molecules within exosomes (e.g., carbonic anhydrase). Clinical studies are also needed to validate the existing data in a broader range of body fluids, with considerable advantages for patients by avoiding or limiting unnecessary invasive procedures and hopefully significantly reducing public health costs. In this sense, the data from studies performed in the urines of patients look very promising [25,26,64,80,81,82,83]. Most of all, we need to tidy up the increasing amount of clinical and pre-clinical data supporting the use of exosomes as a source of tumor biomarkers, using too often different technologies and different ways to obtain exosomes from other body fluids [84,85,86,87,88,89,90,91,92,93,94,95,96,97,98,99,100,101,102,103,104,105,106,107,108,109,110,111]. This review asks for a more strategic approach to obtaining data on exosomes from clinical samples of tumor patients. As challenging news, it has been recently reported that exosomes may deliver therapeutic antibodies that have been shown to maintain their full activity when expressed on exosomes [112]. This finding might be of paramount importance not only for therapeutic use but also for its potential as a new family of biomarkers for both the diagnosis and prognosis of cancer patients. Table 2 summarizes the ongoing clinical trials using exosomes as diagnostic/prognostic tumor biomarkers. It is straightforward from the table that the number of clinical trials is increasing, and the current number is awe-inspiring, up to 65. This means that in the following years, we will have more data to reason about the future directions of clinical research on exosomes. The current clinical research covers a broad panel of exosome-associated potential tumor biomarkers that will hopefully represent a promising future for clinical oncology.

**Table 1 cancers-15-02878-t001:** Data from clinical investigations on extracellular vesicles.

Tumor	Biomarkers	Source	References
**Breast cancer**	Breast cancer resistance protein (BCRP)	Plasma	[113]
	Her2	Plasma Serum	[114,115]
	Glypican-1	Serum	[56]
	Fibronectin	Plasma	[116]
	Periostin	Plasma	[117]
	Del-1	Plasma	[118,119]
	miR-101, miR-372, and miR-373	Serum	[84]
	miR-1246 and miR-21	Plasma	[85]
**Colorectal cancer**	Hsp60	Plasma	[38]
	TSAP6/CEA	Plasma	[86]
	Glypican-1	Plasma	[28]
	CEA	Serum	[45,87]
	CD147	Serum	[87]
		Plasma	[89]
	let-7a, miR-1229, miR-1246, miR-150, miR-21, miR-223, and miR-23a	Serum	[90]
	miR-19	Serum	[91]
	miR-4772-3p	Serum	[92]
	miR-21	Serum	[87]
	miR-221	Serum	[94]
**Esophageal squamous sell sarcinoma**	miR-21	Serum	[95]
**Gastric cancer**	GKN1	Serum	[96]
	TGF-β1	Plasma	[97]
	RNA	Bile	[98]
	miR-423-5p	Serum	[99]
**Hematological tumors**	CD9, CD13, CD19, CD30, CD38, and CD63	Serum	[100]
**Hepatocellular carcinoma**	miR-18a, miR-221, miR-222, and miR-224	Serum	[101]
	miR-718	Serum	[102]
**Laryngeal squamous cell carcinoma**	miR-21 and HOTAIR (lncRNA)	Serum	[103]
**Lung cancer**	NY-ESO-1	Plasma	[104]
	miR-125a-5p, miR-145, and miR-146a	Serum	[105]
	miR-151a-5p, miR-30a-3p, miR-200b-5p, miR-629, miR-100, and miR-154-3p	Plasma	[106]
**Melanoma**	Caveolin-1	Plasma	[24]
	HSP70 and HSP90	Plasma	[120]
	MIA and S100B	Serum	[121]
**Oral squamous cell carcinoma**	CAV-1	Plasma	[43]
**Ovarian cancer**	EpCAM, CD24, and CA-125	Plasma	[122,123,124]
	TGF-beta1 and MAGE3/6,	Plasma	[125]
	miR-21, miR-214, miR-200a, miR-200b, miR-200c, miR-203, miR-205, and miR-141	Serum	[126]
	miR-21, miR-100, miR-200, miR-320, and miR373	Serum	[107]
**Pancreatic cancer**	CD44v6, Tspan 8, EpCAM, and CD104 miR-1246 miR-3976 miR-4306 miR-4644	Serum Urine	[64]
	KRAS P53 mutations	Serum	[71]
	miR-17-5p and miR-21	Serum	[108]
	miR-10b, miR-21, miR-30c, miR-181a, and miR-let7a	Serum	[127]
	Glypican-1	Plasma	[109]
	miR-191, miR-21, and miR-451a	Serum	[110]
	miR-451a	Plasma	[111]
**Prostate cancer (PCa)**	PSA	Plasma	[44,53]
		Urine	[25]
	CA IX	Plasma	[47]
	Survivin	Plasma	[128]
	Exosome levels	Plasma	[41]
	PTEN	Plasma	[129]
	miR-141 and miR-375	Serum	[130]
	miR-1290 and miR-375	Plasma	[131]
	miR-141	Serum	[132]

**Table 2 cancers-15-02878-t002:** Ongoing clinical trials using exosomes in tumor diagnosis.

NCT Number	Status	Disease	Characteristics	Ref.
NCT03235687	Active, not reciting	Prostate Cancer	***Year:*** 2017 ***Population:*** n = 1000; Age: 50 years and older; Sex: male ***Phase:*** Not applicable	[133]
NCT03974204	Withdrawn	Breast Cancer Leptomeningeal Metastasis	***Year:*** 2019 ***Population:*** n = 0; Age: 18 years and older; Sex: female ***Phase:*** Not applicable	[134]
NCT05286684	Recruiting	Breast Cancer	***Year:*** 2023 ***Population:*** n = 30; Age: 18 years and older; Sex: female ***Phase:*** Not applicable	[135]
NCT04781062	Active, not recruiting	Breast Cancer	***Year:*** 2021 ***Population:*** n = 367; Age: 18 years and older; Sex: female ***Phase:*** Not applicable	[136]
NCT02662621	Completed	Cancer (Solid Tumors)	***Year:*** 2015 ***Population:*** n = 71; Age: 18 years and older; Sex: all ***Phase:*** Not applicable	[137]
NCT04530890	Recruiting	Breast Cancer Digestive Cancer Gynecologic Cancer Circulating Tumor DNA Exosomes	***Year:*** 2021 ***Population:*** n = 1000; Age: 18 years and older; Sex: all ***Phase:*** Not applicable	[138]
NCT04258735	Recruiting	Metastatic Breast Cancer	***Year:*** 2019 ***Population:*** n = 300; Age: 18 years and older; Sex: all ***Phase:*** Not applicable	[139]
NCT04556916	Recruiting	Prostate Cancer	***Year:*** 2021 ***Population:*** n = 320; Age: 40 years and older; Sex: male ***Phase:*** Not applicable	[140]
NCT03711890	Recruiting	Pancreatic Carcinoma Pancreatic Intraductal Papillary Mucinous Neoplasm, Pancreatobiliary Type	***Year:*** 2019 ***Population:*** n = 75; Age: 18 years and older; Sex: all ***Phase:*** Not applicable	[141]
NCT02507583	Completed	Malignant Glioma Neoplasms	***Year:*** 2015 ***Population:*** n = 33; Age: 18 years and older; Sex: all ***Phase:*** Phase 1	[142]
NCT05218759	Not yet recruiting	Non-Small Cell Lung Cancer	***Year:*** 2022 ***Population:*** n = 30; Age: 18 to 75 years; Sex: all ***Phase:*** Not applicable	[143]
NCT04427475	Unknown status	NSCLC Patients	***Year:*** 2020 ***Population:*** n = 200; Age: 18 years and older; Sex: all ***Phase:*** Not applicable	[144]
NCT04636788	Unknown status	Pancreas Adenocarcinoma	***Year:*** 2020 ***Population:*** n = 102; Age: 18 years and older; Sex: all ***Phase:*** Not applicable	[145]
NCT03542253	Unknown status	Early Lung Cancer	***Year:*** 2018 ***Population:*** n = 80; Age: child, adult, and older adult; Sex: all Phase: not reported	[146]
NCT04529915	Active, not recruiting	Lung Cancer	***Year:*** 2020 ***Population:*** n = 470; Age: 40 years and older; Sex: all ***Phase:*** not reported	[147]
NCT03821909	Unknown status	Pancreatic Cancer	***Year:*** 2018 ***Population:*** n = 30; Age: 18 to 80 years; Sex: all ***Phase:*** not repoted	[148]
NCT03830619	Completed	Lung Cancer (Diagnosis)	***Year:*** 2017 ***Population:*** n = 1000; Age: 18 to 75 years; Sex: all Phase: not reported	[149]
NCT04394572	Completed	Colorectal Cancer	***Year:*** 2021 ***Population:*** n = 80; Age: 18 years and older; ***Sex:*** all ***Phase:*** not reported	[150]
NCT04155359	Recruiting	Bladder Cancer	***Year:*** 2020 ***Population:*** n = 3000; Age: 45 to 85 years; Sex: all ***Phase:*** not reported	[151]
NCT01344109	Withdrawn	Breast Neoplasms	***Year:*** 2011 ***Population:*** n = 0; Age: 18 years and older; Sex: female ***Phase:*** not reported	[152]
NCT05587114	Recruiting	Lung Cancer Diagnosis	***Year:*** 2022 ***Population:*** n = 150; Age: 40 years and older; Sex: all ***Phase:*** not reported	[153]
NCT05270174	Not yet recruiting	Explore Whether lncRNA-ElNAT1 in Urine Exosomes Can be Used as a New Target for Preoperative Diagnosis of Lymph Node Metastasis	***Year:*** 2023 ***Population:*** n = 75; Age: 18 years and older; Sex: all ***Phase:*** not reported	[154]
NCT03032913	Completed	Pancreatic Ductal Adenocarcinoma (PDAC)	***Year:*** 2017 ***Population:*** n = 52; Age: 18 years and older; Sex: all ***Phase:*** not reported	[155]
NCT02702856	Completed	Prostate Cancer	***Year:*** 2014 ***Population:*** n = 2000; Age: 50 years and older; Sex: male ***Phase:*** not reported	[156]
NCT04523389	Unknown status	Colorectal Cancer	***Year:*** 2020 ***Population:*** n = 172; Age: 18 years and older; Sex: all ***Phase:*** not reported	[157]
NCT03694483	Suspended	Prostate Cancer	***Year:*** 2018 ***Population:*** n = 600; Age: 18 years and older; Sex: male ***Phase:*** not reported	[158]
NCT04661176	Active, not recruiting	Prostate Cancer	***Year:*** 2020 ***Population:*** n = 500; Age: 22 years and older; Sex: male ***Phase:*** not reported	[159]
NCT02393703	Recruiting	Pancreatic Cancer Benign Pancreatic Disease	***Year:*** 2015 ***Population:*** n = 111; Age: 18 years and older; Sex: all ***Phase:*** not reported	[160]
NCT01779583	Unknown status	Gastric Cancer	***Year:*** 2013 ***Population:*** n = 80; Age: 18 years and older; Sex: all ***Phase:*** not reported	[161]
NCT04081194	Unknown status	New Tumor Diagnostics From Human Plasma Samples	***Year:*** 2016 ***Population:*** n = 15; Age: 50 to 90 years; Sex: all ***Phase:*** not reported	[162]
NCT03236688	Suspended	Metastatic Castrate-Resistant Prostate Cancer	***Year:*** 2016 ***Population:*** n = 30; Age: 18 years and older; Sex: male ***Phase:*** not reported	[163]
NCT04629079	Recruiting	Lung Cancer	***Year:*** 2020 ***Population:*** n = 800; Age: 18 years and older; Sex: all ***Phase:*** not reported	[164]
NCT04939324	Active, not recruiting	Lung Cancer Exosomes Non-Small Cell Lung Cancer	***Year:*** 2021 ***Population:*** n = 30; Age: 18 years and older; Sex: all ***Phase:*** Not Applicable	[165]
NCT04288141	Recruiting	HER2-positive Breast Cancer	***Year:*** 2019 ***Population:*** n = 40; Age: 18 years and older; Sex: all ***Phase:*** not reported	[166]
NCT03874559	Unknown status	Rectal Cancer	***Year:*** 2018 ***Population:*** n = 30; Age: 18 years and older; Sex: all ***Phase:*** not reported	[167]
NCT03738319	Unknown status	High-Grade Serous Carcinoma Ovarian Cancer Exosomes Prognosis Early Diagnosis	***Year:*** 2018 ***Population:*** n = 160; Age: 18 years and older; Sex: female ***Phase:*** not reported	[168]
NCT04720599	Completed	Urologic Cancer	***Year:*** 2020 ***Population:*** n = 120; Age: 50 years and older; Sex: male ***Phase:*** not reported	[169]
NCT05101655	Completed	Osteosarcoma Pulmonary Metastases	***Year:*** 2020 ***Population:*** n = 60; Age: 12 to 60 years; Sex: all ***Phase:*** not reported	[170]
NCT04315753	Unknown status	Lung Cancer	***Year:*** 2018 ***Population:*** n = 2000; Age: 55 years and older; Sex: all ***Phase:*** not reported	[171]
NCT03895216	Completed	Bone Metastases	***Year:*** 2018 ***Population:*** n = 34; Age: 18 years and older; Sex: all ***Phase:*** not reported	[172]
NCT04960956	Terminated	Prostate Cancer Urothelial Carcinoma	***Year:*** 2016 ***Population:*** n = 13; Age: 18 years and older; Sex: male ***Phase:*** not reported	[173]
NCT03911999	Completed	Prostate Cancer	***Year:*** 2018 ***Population:*** n = 180; Age: 45 years and older; Sex: male ***Phase:*** not reported	[174]
NCT05572099	Recruiting	Prostate Cancer	***Year:*** 2018 ***Population:*** n = 750; Age: 45 years and older; Sex: male ***Phase:*** not reported	[175]
NCT04323579	Unknown status	Lung Cancer	***Year:*** 2018 ***Population:*** n = 2000; Age: 55 years and older; Sex: all ***Phase:*** not reported	[176]
NCT04357717	Terminated	Prostate Cancer	***Year:*** 2020 ***Population:*** n = 150; Age: 50 years and older; Sex: male ***Phase:*** not reported	[177]
NCT04100811	Recruiting	Prostate Cancer	***Year:*** 2020 ***Population:*** n = 4000; Age: 45 years and older; Sex: male ***Phase:*** not reported	[178]
NCT05463107	Not yet recruiting	Thyroid Cancer Follicular Thyroid Cancer	***Year:*** 2022 ***Population:*** n = 50; Age: 20 to 80 years; Sex: all ***Phase:*** not reported	[179]
NCT04653740	Recruiting	Advanced Breast Cancer	***Year:*** 2020 ***Population:*** n = 25; Age: 18 years and older; Sex: female ***Phase:*** Not applicable	[180]
NCT02147418	Recruiting	Oropharyngeal Cancer	***Year:*** 2015 ***Population:*** n = 30; Age: 18 years and older; Sex: all ***Phase:*** Not reported	[181]
NCT03432806	Recruiting	Colon Cancer Liver Tumors	***Year:*** 2017 ***Population:*** n = 80; Age: 18 years and older; Sex: all ***Phase:*** Not reported	[182]
NCT05397548	Recruiting	Gastric Cancer	***Year:*** 2022 ***Population:*** n = 700; Age: 18 to 80 years; Sex: all ***Phase:*** Not reported	[183]
NCT03811600	Completed	Sleep Apnea Syndromes, Obstructive Cancer	***Year:*** 2019 ***Population:*** n = 90; Age: 18 years and older; Sex: all ***Phase:*** not reported	[184]
NCT03108677	Active, not recruiting	Lung Metastases Osteosarcoma	***Year:*** 2017 ***Population:*** n = 90; Age: 12 to 60 years; Sex: all ***Phase:*** not reported	[185]
NCT04499794	Recruiting	Untreated Advanced NSCLC Patients FISH-Identified ALK Fusion (Positive or Negative)	***Year:*** 2020 ***Population:*** n = 75; Age: 18 years and older; Sex: all ***Phase:*** not reported	[186]
NCT04182893	Unknown status	Pulmonary Nodules	***Year:*** 2019 ***Population:*** n = 400; Age: 18 years and older; Sex: all ***Phase:*** not reported	[187]
NCT02464930	Unknown status	Barrett’s Esophagus Gastroesophageal Reflux Esophageal Adenocarcinoma	***Year:*** 2015 ***Population:*** n = 220; Age: 18 years and older; Sex: all ***Phase:*** not reported	[188]
NCT05625529	Not yet recruiting	Pancreas Cancer Exosomes Extracellular Vesicles Pancreatic Neoplasms	***Year:*** 2022 ***Population:*** n = 1000; Age: 18 years and older; Sex: all ***Phase:*** not reported	[189]
NCT03581435	Unknown status	Proteinosis Gallbladder Carcinoma	***Year:*** 2018 ***Population:*** n = 50; Age: 18 years and older; Sex: all ***Phase:*** not reported	[190]
NCT03102268	Unknown status	Cholangiocarcinoma Benign Biliary Stricture	***Year:*** 2017 ***Population:*** n = 80; Age: 18 years and older; Sex: all ***Phase:*** not reported	[191]
NCT05705583	Recruiting	Renal Cell Carcinoma	***Year:*** 2023 ***Population:*** n = 100; Age: 18 years and older; Sex: all ***Phase:*** not reported	[192]
NCT03334708	Recruiting	Pancreatic Cancer Pancreatic Diseases Pancreatitis Pancreatic Cyst	***Year:*** 2017 ***Population:*** n = 700; Age: 18 years and older; Sex: all ***Phase:*** not reported	[193]
NCT03800121	Recruiting	Sarcoma	***Year:*** 2018 ***Population:*** n = 30; Age: 18 years and older; Sex: all ***Phase:*** not reported	[194]
NCT05744076	Active, not recruiting	Melanoma	***Year:*** 2019 ***Population:*** n = 150; Age: 18 years and older; Sex: all ***Phase:*** not reported	[195]
NCT04053855	Recruiting	Clear Cell Renal Cell Carcinoma	***Year:*** 2020 ***Population:*** n = 100; Age: 18 years and older; Sex: all ***Phase:*** not reported	[196]

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
