# Peer review of "The Importance of Detecting, Quantifying, and Characterizing Exosomes as a New Diagnostic/Prognostic Approach for Tumor Patients"

_cancers, 2023, doi:10.3390/cancers15112878_

Round 1

Reviewer 1 Report

Logozzi et al. reported the importance of exosome characterization in clinical samples of tumor patients highlighting that plasmatic exosome amount is higher in patients than in controls and that express tumor markers (e.g., PSA and CEA).

This is an interesting review; it could be improved according to following suggestions.

Major revision:

1.      Line 100-101: The authors should revise the sentence reporting that “NTA analysis can covers a broad range of particles size, ranging from 10-30nm to 400-500….” or insert references that validate this information. NTA is not capable of detecting particles smaller than about 30 nm, while it can detect particles up to 800nm.

2.      The authors reported The last accepted guidelines (MISEV2018) have agreed on the following points that are required to establish that our sample actually contains exosomes: (i) the presence of at least one transmembrane protein associated with the exosomal plasma membrane (e.g., tetraspanins CD9, CD63, CD81); (ii) the presence of a cytosolic protein (e.g., TSG101, ALIX) “. This sentence and in general some part of paper that emphasize a specific identification for exosome markers should be revisited/remodulated. In fact, MISEV 2018 reported “Although numerous proteomic analyses have highlighted proteins commonly found in exosome preparations, it is becoming clear that these do not represent “exosome-specific” markers but rather “exosome-enriched” proteins, as different subsets of secreted EVs contain many common markers.

3.      Regard clinical studies would be appreciated a table reporting the ongoing and complete clinical trials on EVs /exosomes and same information about them. This information could be inserted in the paragraph 3 “A role of exosome in cancer……”

4.      Regard the microRNA analysis and relative problems (such as co-isolation of miRNA with lipoproteins or other proteins lines 208-219): the authors could insert some sentences about other EV/exosome isolation methods in addition to immunocapture-based methodology that resolved this problem (e.g., some type of bench centrifuge step and others). It would be appreciated.

5.      The authors should include some other information on the presence of DNA in the EVs/exosomes (lines 223-226) reporting some recent references.

6.      Table 1 is appreciated and useful but must be enriched by other information such as i) purification methods of EVs; ii) methodology for both miRNA and antigen analysis; iii) patient number and healthy subjects enrolled in every study; iv) microRNAs in hematological malignancies, melanoma oral squamous cell carcinoma.

7.      Check the references in the text and in the table 1: e.g., reference 90, specifically, in the table does not match with reported data and consequently also the others to follow.

Minor revisions:

1.      Line 64:  to replace plasma with blood and insert comma after urines.

2.      Line 78: to eliminate “the s”.

3.      Line 197: to replace “RNAs” with “nucleic acids”.

Minor editing of English language required

Author Response

Reviewer 1

Logozzi et al. reported the importance of exosome characterization in clinical samples of tumor patients highlighting that plasmatic exosome amount is higher in patients than in controls and that express tumor markers (e.g., PSA and CEA).

This is an interesting review; it could be improved according to following suggestions.

First of all, we sincerely thank the reviewer for the efforts done to improve our manuscript that led to a real improvement of both the script and the level information of the revised version. It was very much appreciated.

Major revision:

  1. Line 100-101: The authors should revise the sentence reporting that “NTA analysis can covers a broad range of particles size, ranging from 10-30nm to 400-500….” or insert references that validate this information. NTA is not capable of detecting particles smaller than about 30 nm, while it can detect particles up to 800nm.

We have revised the text accordingly

  1. The authors reported “The last accepted guidelines (MISEV2018) have agreed on the following points that are required to establish that our sample actually contains exosomes: (i) the presence of at least one transmembrane protein associated with the exosomal plasma membrane (e.g., tetraspanins CD9, CD63, CD81); (ii) the presence of a cytosolic protein (e.g., TSG101, ALIX) “. This sentence and in general some part of paper that emphasize a specific identification for exosome markersshould be revisited/remodulated. In fact, MISEV 2018 reported “Although numerous proteomic analyses have highlighted proteins commonly found in exosome preparations, it is becoming clear that these do not represent “exosome-specific” markers but rather “exosome-enriched” proteins, as different subsets of secreted EVs contain many common markers.
  2. We have revised the text accordingly
  3. Regard clinical studies would be appreciated a table reporting the ongoing and complete clinical trials on EVs /exosomes and same information about them. This information could be inserted in the paragraph 3 “A role of exosome in cancer……”

A new table listing the ongoing clinical trials in the tumor diagnostics area has been included in the revision

  1. Regard the microRNA analysis and relative problems (such as co-isolation of miRNA with lipoproteins or other proteins lines 208-219): the authors could insert some sentences about other EV/exosome isolation methods in addition to immunocapture-based methodology that resolved this problem (e.g., some type of bench centrifuge stepand others). It would be appreciated.

New sentences have been included accordingly

  1. The authors should include some other information on the presence of DNA in the EVs/exosomes (lines 223-226) reporting some recent references.

New references and discussion have been included

  1. Table 1 is appreciated and useful but must be enriched by other information such as i)purification methods of EVs; ii) methodology for both miRNA and antigen analysis; iii) patient number and healthy subjects enrolled in every study; iv) microRNAs in hematological malignancies, melanoma oral squamous cell carcinoma.

We share with the reviewer the need for additional information in table 1. However,when we tried to fill up the table with other information the result was rambling and confusing for different reasons. First of all very few studies actually contained the new information and the table resulted too huge and with many missing informations. Thus , we decided to maintain the same structure of the table but carefully checking the consistency of the reported references for each study. In considering the new table 2 we retained that the overall information on the results of the clinical studies together with the ongoing clinical studies will offer to the readers a unique sight on what clinical research on the diagnostic use of exosome in cancer patients has offered and will offer in the next few years

  1. Check the references in the text and in the table 1: e.g., reference 90, specifically, in the table does not match with reported data and consequently also the others to follow.

WE HAVE CORRECTED THE SOURCE OF THE EXOSOMES IN THE REF 90, THAT WA BILE ACTUALLY, BUT THE FOLLOWING REFERENCES ARE CORRECT INDEED.

References have been carefully checked

Minor revisions:

  1. Line 64:  to replace plasma with blood and insert comma after urines.
  2. Line 78: to eliminate “the s”.
  3. Line 197: to replace “RNAs” with “nucleic acids”.

All the requested minor changes have been done

Reviewer 2 Report

The manuscript is clearly written and aims at summarizing the technical strategies to detect and quantify EVs in patients' fluids.

The cited literature seems updated (rarely earlier than 2017). However, most of it refers to other review articles, rather than data-rich research manuscripts. 

There are multiple typos and word iterations.

The authors are encouraged to carefully proofread the text. 

Author Response

Reviewer  2

The manuscript is clearly written and aims at summarizing the technical strategies to detect and quantify EVs in patients' fluids.

The cited literature seems updated (rarely earlier than 2017). However, most of it refers to other review articles, rather than data-rich research manuscripts. 

Comments on the Quality of English Language

There are multiple typos and word iterations.

The authors are encouraged to carefully proofread the text. 

We thank the reviewer for the encouraging comments and advices. We have carefully checked the whole manuscript for typos and tautologies and updated the ref list with new quotations.

Reviewer 3 Report

This manuscript analyzes some interesting aspects, but there are several areas that need to be emphasized and elaborated before it is accepted for publication.

Sometimes, the language used is far too colloquial (e.g.”your sample”, "our samples", “allowed us”). Appropriate language should be used.

Author Response

Reviewer 3

The article you submitted to me (“The importance to detect, quantify and characterize exosomes in clinical samples of tumor patients”) is interesting, but in order to publish it, a thorough revision is needed. The manuscript entitled “The importance to detect, quantify and characterize exosomes in clinical samples of tumor patients” discusses the utility of exosomes as novel biomarkers in cancer patients. This manuscript analyzes some interesting aspects, but there are several areas that need to be emphasized and elaborated before it is accepted for publication.

First of all, we sincerely thank the reviewer for the efforts done to improve our manuscript that led to a real improvement of both the script and the level information of the revised version. It was very much appreciated.

Major comments:

Section 1:

Lines 40-42: review this sentence.

For example: (i) EVs range in size from 30nm to 1 μm; (ii) the range from 200nm to 1 μm identifies microvesicles and the range from 30 nm to 150nm identifies exosomes.

Lines 44-45: “but other 44 mechanisms cannot be excluded and are currently under investigation worldwide”, please add a reference or removing this sentence. Line 53: please add a reference.

Lines 54-60: in these sentences are described only exosomes, I suggest reviewing this paragraph removing EVs and citing only exosomes.

Lines 74-83: this paragraph is not clear. I suggest writing short sentences and emphasizing important aspects.

We have carefully revised the text accordingly with the reviewer’s comments and advice

Section 2: Sometimes, the language used is far too colloquial (e.g.”your sample”, "our samples", “allowed us”). Appropriate language should be used.

Line 95: “in human body fluids”, I suggest to remove this sentence.

 Lines 104-112: I suggest simplifying the paragraph and emphasizing the use of TEM for qualitative analysis. It is difficult to understand the whole argument. I also suggest replacing "TEM analysis can’t be avoided" with "it is advisable to evaluate exosomes by TEM analysis."

Line 113: replace “EVs” with “exosomes” I suggest removing “Moreover it is expensive in terms of both the volumes required for the analysis and the time needed to obtain the results; it is undoubtfully a qualitative analysis allowing to the detect many proteins at the same time” (lines 120-122) and “Fluorescence Activated Cell Sorter (FACS), while equipped with a nanoscale (nanoscale flow cytometry), does not allow a broad spectrum of analysis, and simultaneous analysis of different samples” (lines 135-137).

We have carefully revised the text accordingly with the reviewer’s comments and advice, most of all avoiding all the too colloquial sentences

Sections 3, 4 and 5: Expand these sections.

We have expanded these sections accordingly

Section 6: Conclusions section will be reviewed during the second revision. Minor comments: Please fix the typos and grammatical errors and specify acronyms the first time they are used

Conclusions has been expanded and revised accordingly, also by including a new table reporting the ongoing clinical trials on the used of exosomes in the diagnostic of tumor patients (Table 2)

Reviewer 4 Report

This manuscript is interesting. In this review, the authors summary that “The importance of detect, quantify, and characterize exosomes in clinical samples from tumor patients”. However,the evidences to illustrate the title are insufficient and the organization of the whole text need to be improved. In general, this article is more like a popular science text than an review. I suggest that the authors rewrite this review. Here are my suggestions.

1.The manuscript needs to be carefully edited by someone with expertise in technical English editing, paying particular attention to English grammar, spelling and sentence structure so that the study is clear to the readers. For example, “The technical approach to obtain EVs includes methods to physically purify EVs, n and methods to characterize EVs, such as Nanosight Tracking Analysis (NTA), immunocapture-based ELISA and nano-scale flow cytometry” (in Page 1, line 26-28), There is a grammatical error in this sentence; More-over, Immunocapture-based ELISA allowed us to quantify and characterize several clini-cal samples simultaneously and in a broader population of EVs as compared with NFC [53,54], method described in [55]. (in Page 4, line 151-154). There is a grammatical error in this sentence.

2. The organization of the abstract is not clear, please state the main points to be summarized in the abstract.

3. The topic is broad and unfocused, please write the title according to the main points summarized in this manuscript.

4. If abbreviation had been defined in the text when used for the first time, abbreviation is recommended in the text below. For example, line 95 “Nanoparticle Tracking Analysis (NTA)”.

5. In Introduction, there is a lot about EVS. I suggest focusing on exosomes rather than EVS. In this section, I suggest adding a schematic diagram to explain the structure and the production mechanism of exosomes and so on.

6. In the section of “A technical insight”, the techniques used to purify and characterize EVS and exosomes have been described, and the description of EVS is too much. I suggest focusing on exosomes.

7. In the section of “A role of exosomes in cancer from preclinical to clinical data,I suggest that preclinical and clinical data should be classified by tumor type. This suggestion also applies to the section of “4. Exosomes deliver enzymatic activity” and “5. Exosomes deliver RNAs”. I also suggest adding tables to show the above.

“The technical approach to obtain EVs includes methods to physically purify EVs, n and methods to characterize EVs, such as Nanosight Tracking Analysis (NTA), immunocapture-based ELISA and nano-scale flow cytometry” (in Page 1, line 26-28), There is a grammatical error in this sentence

More-over, Immunocapture-based ELISA allowed us to quantify and characterize several clini-cal samples simultaneously and in a broader population of EVs as compared with NFC [53,54], method described in [55](in Page 4, line 151-154). There is a grammatical error in this sentence.

Author Response

Reviewer 4

This manuscript is interesting. In this review, the authors summary that “The importance of detect, quantify, and characterize exosomes in clinical samples from tumor patients”. However,the evidences to illustrate the title are insufficient and the organization of the whole text need to be improved. In general, this article is more like a popular science text than an review. I suggest that the authors rewrite this review. Here are my suggestions.

First of all, we sincerely thank the reviewer for the efforts done to improve our manuscript that led to a real improvement of both the script and the level information of the revised version. It was very much appreciated.

1.The manuscript needs to be carefully edited by someone with expertise in technical English editing, paying particular attention to English grammar, spelling and sentence structure so that the study is clear to the readers.

For example, “The technical approach to obtain EVs includes methods to physically purify EVs, n and methods to characterize EVs, such as Nanosight Tracking Analysis (NTA), immunocapture-based ELISA and nano-scale flow cytometry” (in Page 1, line 26-28),

There is a grammatical error in this sentence; “More-over, Immunocapture-based ELISA allowed us to quantify and characterize several clini-cal samples simultaneously and in a broader population of EVs as compared with NFC [53,54], method described in [55]”. (in Page 4, line 151-154). There is a grammatical error in this sentence.

We have carefully revised the text for the English grammar, spelling and sentence structure, also considering those emphasized by the reviewer.

  1. The organization of the abstract is not clear, please state the main points to be summarized in the abstract.

We have revised the abstract accordingly

  1. The topic is broad and unfocused, please write the title according to the main points summarized in this manuscript.

We have changed the title accordingly

  1. If abbreviation had been defined in the text when used for the first time, abbreviation is recommended in the text below. For example, line 95 “Nanoparticle Tracking Analysis (NTA)”.

We have used the abbreviations accordingly

  1. In Introduction, there is a lot about EVS. I suggest focusing on exosomes rather than EVS. In this section, I suggest adding a schematic diagram to explain the structure and the production mechanism of exosomes and so on.

We have changed all the EVs in exosomes as appropriate

  1. In the section of “A technical insight”, the techniques used to purify and characterize EVS and exosomes have been described, and the description of EVS is too much. I suggest focusing on exosomes.

We have focused this section on exosomes accordingly with the reviewer’s suggestion

  1. In the section of “A role of exosomes in cancer from preclinical to clinical data”,I suggest that preclinical and clinical data should be classified by tumor type. This suggestion also applies to the section of “4. Exosomes deliver enzymatic activity” and “5. Exosomes deliver RNAs”. I also suggest adding tables to show the above.

We added a new table on the ongoing clinical trials (Table 2)

Comments on the Quality of English Language

“The technical approach to obtain EVs includes methods to physically purify EVs, n and methods to characterize EVs, such as Nanosight Tracking Analysis (NTA), immunocapture-based ELISA and nano-scale flow cytometry” (in Page 1, line 26-28), There is a grammatical error in this sentence; 

“More-over, Immunocapture-based ELISA allowed us to quantify and characterize several clini-cal samples simultaneously and in a broader population of EVs as compared with NFC [53,54], method described in [55]”. (in Page 4, line 151-154). There is a grammatical error in this sentence.

The English has been carefully revised by a mother tongue person

Round 2

Reviewer 4 Report

This manuscript can be accepted in present form.